# High Incidence Rate of SARS-CoV-2 Infection in Health Care Workers at a Dedicated COVID-19 Hospital: Experiences of the Pandemic from a Large Mexican Hospital

**DOI:** 10.3390/healthcare10050896

**Published:** 2022-05-12

**Authors:** Nallely Bueno-Hernández, José Damian Carrillo-Ruíz, Lucía A. Méndez-García, Salma A. Rizo-Téllez, Rebeca Viurcos-Sanabria, Alisson Santoyo-Chávez, René Márquez-Franco, Alejandro Aguado-García, Neyla Baltazar-López, Yoshio Tomita-Cruz, Eira Valeria Barrón, Ana Laura Sánchez, Edna Márquez, Ruben Fossion, Ana Leonor Rivera, Luis Ruelas, Octavio A. Lecona, Gustavo Martínez-Mekler, Markus Müller, América G. Arroyo-Valerio, Galileo Escobedo

**Affiliations:** 1Laboratory of Proteomics, Research Division, General Hospital of Mexico “Dr. Eduardo Liceaga”, Mexico City 06726, Mexico; nallely_bh5@yahoo.com.mx (N.B.-H.); santoyoalison@gmail.com (A.S.-C.); 2Research Directorate, General Hospital of Mexico “Dr. Eduardo Liceaga”, Mexico City 06726, Mexico; josecarrilloruiz@yahoo.com (J.D.C.-R.); rene.marquez@c3.unam.mx (R.M.-F.); ane_sito@yahoo.com.mx (A.A.-G.); neylabaltazar@yahoo.com.mx (N.B.-L.); dryoshiotomita@gmail.com (Y.T.-C.); 3Department of Neurology and Neurosurgery, General Hospital of Mexico “Dr. Eduardo Liceaga”, Mexico City 06726, Mexico; 4Facultad de Ciencias de la Salud, Universidad Anáhuac, Huixquilucan 52786, Mexico; 5Laboratory of Immunometabolism, Research Division, General Hospital of Mexico “Dr. Eduardo Liceaga”, Mexico City 06726, Mexico; angelica.mendez.86@hotmail.com (L.A.M.-G.); sart.17.04@gmail.com (S.A.R.-T.); viurcos.reb@hotmail.com (R.V.-S.); 6Unidad de Medicina Genómica, General Hospital of Mexico “Dr. Eduardo Liceaga”, Mexico City 06726, Mexico; valeirabarron@gmail.com (E.V.B.); alss_7@yahoo.com.mx (A.L.S.); cednam@gmail.com (E.M.); 7Instituto de Ciencias Nucleares, Universidad Nacional Autónoma de México, Mexico City 04510, Mexico; ruben.fossion@gmail.com (R.F.); ana.rivera@nucleares.unam.mx (A.L.R.); 8Centro de Ciencias de la Complejidad (C3), Universidad Nacional Autónoma de México, Mexico City 04510, Mexico; luise.ruelasz@gmail.com (L.R.); tavolecona@ciencias.unam.mx (O.A.L.); 9Instituto de Ciencias Físicas, Universidad Nacional Autónoma de México, Mexico City 04510, Mexico; mekler@icf.unam.mx; 10Instituto de Investigación en Ciencias Básicas y Aplicadas, Universidad Autónoma del Estado de Morelos, Cuernavaca 62209, Mexico; muellerm@uaem.mx

**Keywords:** health care workers, SARS-CoV-2, COVID-19, incidence of infection, IgG antibody, pandemic

## Abstract

Health care workers (HCW) are at high risk of severe acute respiratory syndrome coronavirus 2 (SARS-CoV-2) infection. The incidence of SARS-CoV-2 infection in HCW has been examined in cross-sectional studies by quantitative polymerase chain reaction (qPCR) tests, which may lead to underestimating exact incidence rates. We thus investigated the incidence of SARS-CoV-2 infection in a group of HCW at a dedicated coronavirus disease 2019 (COVID-19) hospital in a six-month follow-up period. We conducted a prospective cohort study on 109 participants of both sexes working in areas of high, moderate, and low SARS-CoV-2 exposure. qPCR tests in nasopharyngeal swabs and anti-SARS-CoV-2 IgG serum antibodies were assessed at the beginning and six months later. Demographic, clinical, and laboratory parameters were analyzed according to IgG seropositivity by paired Student’s T-test or the chi-square test. The incidence rate of SARS-CoV-2 infection was considerably high in our cohort of HCW (58%), among whom 67% were asymptomatic carriers. No baseline risk factors contributed to the infection rate, including the workplace. It is still necessary to increase hospital safety procedures to prevent virus transmissibility from HCW to relatives and non-COVID-19 patients during the upcoming waves of contagion.

## 1. Introduction

The global outbreak of coronavirus disease 2019 (COVID-19) was caused by the novel severe acute respiratory syndrome coronavirus 2 (SARS-CoV-2) [1]. The SARS-CoV-2 is highly transmissible to humans, and health care workers (HCW) attending to COVID-19 patients are at a much higher risk of infection than the general population [2,3]. A study conducted at Rutgers University and two affiliated university hospitals reported that the prevalence of SARS-CoV-2 infection is dramatically higher in HCW than in non-health care professionals (7.3% versus 0.4%, respectively) [4]. A cross-sectional study conducted at the Hospital Universitario of Fuenlabrada in Spain informed a SARS-CoV-2 infection rate of 19.9% in HCW [5]. Kenan Rodriguez and colleagues conducted a cross-sectional descriptive study in two health departments belonging to the Spanish National Public Health System, finding an incidence of SARS-CoV-2 infection of 9.7% in HCW [6]. Likewise, another study reported a SARS-CoV-2 infection prevalence of 14.8% in HCW employed in a large tertiary community hospital in New York City [7]. This information suggests that the SARS-CoV-2 infection rate appears reasonably low in health care personnel who faced at least two COVID-19 waves. However, most reports examining the incidence of SARS-CoV-2 infection in HCW were conducted in cross-sectional studies using quantitative polymerase chain reaction (qPCR) tests to detect positive cases. Since qPCR amplifies the virus’s genetic material, this methodology can only identify active infection cases, which may lead to underestimating asymptomatic patients and incidence rates among health care professionals [8]. For this reason, we believe the SARS-CoV-2 infection incidence in HCW should be assessed by conducting prospective, longitudinal cohort studies measuring specific IgG antibodies against the virus to detect all personnel exposed to the infection, irrespective of whether they showed symptoms or not.

At the pandemic’s beginning, the Ministry of Health switched the General Hospital of Mexico (GHM) from a tertiary care center to a dedicated COVID-19 hospital. For this reason, we conducted a prospective cohort study to investigate the incidence rate of SARS-CoV-2 infection in HCW of the GHM by examining anti-SARS-CoV-2 IgG antibody serum titers at the beginning of the study and six months later. We also explored risk factors potentially contributing to infection rates in our study cohort, such as comorbidities and working in areas of high, moderate, and low viral exposure.

## 2. Materials and Methods

### 2.1. Trial Design and Ethical Considerations

This prospective cohort study consisted of enrolling doctors, nurses, researchers, clinical lab technicians, psychologists, rehabilitators, and administrative personnel from the GHM, one of the largest hospitals of the Ministry of Health, designated for treating COVID-19 patients in Mexico City. The study was approved by the Ethics and Clinical Research committees of the GHM (Approval No: DI/20/501/04/32) and was conducted in strict adherence to the principles of the 1964 Declaration of Helsinki and its posterior amendment in 2013. The enrollment and follow-up of participants took place from August 2020 to January 2021. This study met the Strengthening the Reporting of Observational Studies in Epidemiology (STROBE) statement guidelines for reporting observational studies.

### 2.2. Selection of Participants

We invited volunteers to participate in the study with no previous diagnosis of SARS-CoV-2, evidenced by a negative quantitative PCR (qPCR) test in oropharyngeal swabs and negative IgG antibodies against the virus, aged 18 years and above, and with no previous COVID-19 vaccination. All HCW that agreed to take part in the study signed an informed consent form and received a full explanation of the purposes and procedures of the study. Personnel developing administrative tasks were considered to have low SARS-CoV-2 exposure within hospital facilities. Personnel handling and processing blood samples without having contact with COVID-19 patients were believed to have moderate SARS-CoV-2 exposure within the hospital. Personnel working in the emergency room and intensive care units were supposed to have high SARS-CoV-2 exposure within hospital facilities. We chose six months to follow up with all participants based on previous studies examining the COVID-19 incidence in HCW [4,9]. All HCW enrolled in the study agreed to receive weekly monitoring via a phone call interview to register any suggestive COVID-19 symptoms and attend at least two appointments for medical consultation, clinical evaluation, and blood sampling. We provided all HCW enrolled in the protocol with the same N95 respirators, disposable surgical gowns, disposable surgical caps, and face shield masks as personal protection equipment (PPE) throughout the study. We also confirmed that all HCW correctly wore PPE independently of working in low, moderate, or high SARS-CoV-2 exposure areas within the hospital. We labeled a participant as positive for SARS-CoV-2 exposure if the IgG test was positive at the end of the study.

### 2.3. Sample Size Estimation

The number of workers employed at the General Hospital of Mexico is approximately seven thousand. At the pandemic’s beginning, two thousand aged workers were instructed to lock down at home. At this time, we initially estimated a sample size considering that the number of active personnel was five thousand. Based on previous studies [4,5], we expected that 13% of workers were at higher risk of SARS-CoV-2 infection (*n* = 650). Using the equation for population proportion *n* = z^2^ (p × q)/e^2^ + (z^2^(p × q))/*N*, with a margin of error of 10% and a confidence level of 99%, we obtained *n* = 84. We considered 25% losses and finally found a sample size of *n* = 105 health care professionals.

### 2.4. Clinical Evaluation of Volunteers

Demographic, anthropometric, clinical, and laboratory parameters were collected from all study participants. The workplace within the hospital and any current symptoms concurring with SARS-CoV-2 infection were also registered. At the beginning of the study, we took from all participants an initial oropharyngeal swab for the qPCR test and a 4 mL blood sample for specific IgG antibody detection of anti-SARS-CoV-2. According to the World Health Organization (WHO) technical guidance [10], when participants described any symptoms concurring with COVID-19, they were subjected to another qPCR test with nasopharyngeal swab for SARS-CoV-2 detection. At the end of the study, all participants agreed to donate another 4 mL blood sample for specific IgG antibody detection.

### 2.5. IgG Seroprevalence Analysis

Specific IgG antibody levels against the SARS-CoV-2 nucleocapsid (N) protein were measured in triplicate by the Enzyme Linked-ImmunoSorbent Assay (ELISA) by a standardized kit from Abcam (Abcam, ab274339, Cambridge, UK), using a microplate reader at 450 nm.

### 2.6. Statistics

We estimated the normality of data by the Shapiro–Wilk test. At the beginning and the end of the study, clinical, anthropometric, and laboratory data were analyzed by paired Student’s T-test. We estimated differences in sex proportion by the chi-square test and considered differences significant when *p* < 0.05. We performed statistical analyses using the SPSS version 22 (IBM, Armonk, NY, USA).

## 3. Results

Figure 1 illustrates the selection process of participants enrolled in the study based on the inclusion and exclusion criteria (Figure 1).

At the end of the study, 64 participants showed specific IgG antibodies against the virus, denoting a 58% incidence of SARS-CoV-2 infection (Table 1). Among IgG seropositive participants, 21 (33%) were symptomatic and presented symptoms concurring with COVID-19, including fever, headache, fatigue, body pain, anosmia, and dysgeusia. Moreover, three (14%) of the symptomatic participants needed non-invasive respiratory support. In parallel, 67% (*n* = 43) of the IgG seropositive participants were asymptomatic carriers and reported no COVID-19 symptoms during the six-month follow-up. All symptomatic participants were confirmed by a qPCR test, and all of them showed serum titers of IgG antibodies anti-SARS-CoV-2 at the end of the study.

At the beginning of the study (baseline), a retrospective analysis revealed no significant differences between participants who further showed positive or negative IgG serum titers for demographic, anthropometric, and laboratory parameters (Table 1). For comorbidities, hypertension was significantly higher in IgG seronegative participants than in IgG seropositive subjects, with no differences in the prevalence of obesity and Diabetes Mellitus (Table 1). Furthermore, there were no differences between IgG seropositive and seronegative participants concerning the correct use of PPE throughout the study (Table 1).

In participants working in hospital areas of low exposure, the incidence of SARS-CoV-2 infection was 50% (Table 1). A similar trend was found in participants working in areas of moderate exposure, wherein the incidence of SARS-CoV-2 infection was 52.5% (Table 1). Conversely, participants working in areas of high viral exposure tended to have the highest incidence of SARS-CoV-2 infection (64.9%) without showing significant differences (Table 1). We found no differences in the correct use of PPE among participants working in low, moderate, or high SARS-CoV-2 exposure areas within the GHM.

## 4. Discussion

Herein, we found that the incidence of SARS-CoV-2 infection was as high as 58% in our study sample of HCW. Most studies examining the incidence of SARS-CoV-2 infection in HCW have reported incidence rates of 5–29% [4,5,11,12], which significantly differ from our results. These apparently controversial findings can be explained since most studies assessing the incidence of SARS-CoV-2 infection use the qPCR test in either cross-sectional or prospective studies, which may lead to underestimating actual incidence rates [13,14]. As far as we know, this is one of the first prospective cohort studies reporting a high incidence of SARS-CoV-2 infection in a cohort of HCW by examining titers of anti-SARS-CoV-2 IgG antibodies in a six-month exposure time. A study conducted on a group of HCW from a tertiary care center in Japan showed that almost half of the overlooked infection cases by the qPCR test could be identified when measuring anti-SARS-CoV-2 serum antibodies [15]. Therefore, we think the incidence of SARS-CoV-2 infection in HCW should be assessed by IgG antibody measurement in prospective cohort studies as a strategy that can help identify the personnel exposed to the virus irrespective of having presented COVID-19 symptoms or not. However, it is feasible that converting the GHM into an entirely dedicated COVID-19 care center could have impacted the SARS-CoV-2 infection rate in our study compared to other reports [4,5,11,12]. Therefore, our findings may not necessarily apply to all health care settings admitting COVID-19 patients.

In line with previous studies [16,17,18], we found that neither demographic and anthropometric characteristics nor laboratory parameters at the beginning of the follow-up were related to the incidence of SARS-CoV-2 infection in HCW of the GHM. However, we observed that hypertension was higher in IgG seronegative participants than in IgG seropositive volunteers. It is feasible that participants recognizing themselves as a vulnerable population with comorbidities such as hypertension had more safety procedures than volunteers without known comorbidities, even though our data show that both groups followed the minimal requirements of PPE.

We found that the workplace within the hospital was not significantly related to the incidence of SARS-CoV-2 infection when all health care professionals were wearing PPE correctly. HCW at the frontline, including emergency room and intensive care units, had similar infection rates to those working in moderate- and low-exposure areas such as hospital administration and medical social work. Our results align with previous studies reporting that the risk of SARS-CoV-2 infection in HCW is not necessarily associated with the workplace but with other factors such as community exposure and contact with family members who are COVID-19 asymptomatic carriers, and the inaccurate use of PPE [17,18].

Finally, our results indicate that only 33% of IgG seropositive participants presented COVID-19 symptoms, which means that two in three subjects were asymptomatic carriers. Abdulla A. Damluji and colleagues found 4.8% of asymptomatic carriers among HCW [19], while another study reported that 40% of HCW with SARS-CoV-2 infection did not develop any symptoms concurring with COVID-19 [20]. Altogether, our protocol shows many SARS-CoV-2 asymptomatic carriers among HCW, which may contribute to increasing transmissibility of the disease within the hospital and in the community.

We are inclined to point out some strengths and limitations to the study. The sample size (*n* = 109) could be a limitation considering that more than seven thousand employees work in the General Hospital of Mexico. However, besides sample size estimation suggesting that 105 participants were enough for statistical analysis, the number of HCW enrolled allowed us to monitor any suggestive COVID-19 symptoms individually for six months, including weekends, strengthening our ability to estimate infection rates and identify asymptomatic carriers. Furthermore, the fact that we conducted this study in one single care center could also be a limitation, considering that a multicenter trial provides an increased number of patients from different geographic places and socioeconomic statuses. Nevertheless, the General Hospital of Mexico is one of the largest tertiary care centers in the country, attending nearly 200,000 low- and middle-income patients from the south-central region of Mexico per year. Therefore, our findings reflect, to some extent, the exposure levels of HCW working in dedicated COVID-19 hospitals in Mexico and may help consider effective strategies to face emerging health threats like SARS-CoV-2.

## 5. Conclusions

This prospective study shows, for the first time, in a cohort of HCW followed-up for six months, that the incidence rate of SARS-CoV-2 infection was as high as 58%, among which 67% were asymptomatic carriers. There were no risk factors contributing to the increase in the infection rate, including the workplace within the hospital. There is still a considerable need to increase safety procedures in COVID-19 hospitals, where HCW exposed to SARS-CoV-2 infection can increase virus transmissibility to other health care professionals, relatives, and patients.

## Figures and Tables

**Figure 1 healthcare-10-00896-f001:**
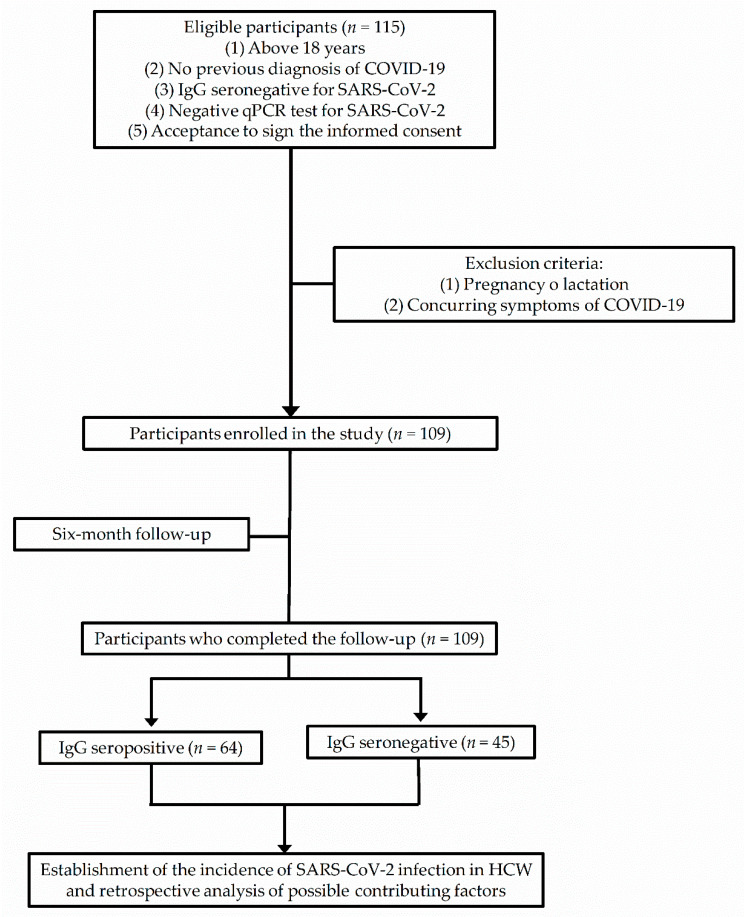
Schematic flow chart showing the selection process of participants enrolled in the study based on the inclusion and exclusion criteria. We conducted the enrollment of participants in strict adherence to the principles of the 1964 Declaration of Helsinki and its posterior amendment in 2013, meeting the STROBE guidelines for reporting observational studies. Abbreviations: IgG, immunoglobulin G; qPCR, quantitative polymerase chain reaction; SARS-CoV-2, severe acute respiratory syndrome coronavirus 2; COVID-19, coronavirus disease 2019; STROBE, Strengthening the Reporting of Observational Studies in Epidemiology; HCW, health care workers.

**Table 1 healthcare-10-00896-t001:** Demographic, anthropometric, clinical, laboratory, and working characteristics of HCW.

Parameter	IgG Seropositive *n* = 64			IgG Seronegative *n* = 45			*P*^A^ vs. ^B^
	Baseline ^A^	Final ^C^	*P*^A^ vs. ^C^	Baseline ^B^	Final ^D^	*P*^B^ vs. ^D^	
*Demographic characteristics*							
Age, years ($\bar{x}$ ± SD)	40 ± 10	-	-	43 ± 11	-	-	0.21
Men, *n* (%)	22 (34)	-	-	14 (31)	-	-	0.83
Women, *n* (%)	42 (66)	-	-	31 (69)	-	-	0.44
*Working exposure level*							
High, *n* (%)	37 (64.9)	-	-	20 (44.4)	-	-	0.19
Moderate, *n* (%)	22 (52.4)	-	-	20 (47.6)	-	-	0.38
Low, *n* (%)	5 (50)	-	-	5 (50)	-	-	0.39
Use of PPE, *n* (%)	60 (94)	-	-	39 (87)	-	-	0.28
*Prevalence of comorbidities*							
Sedentary lifestyle, *n* (%)	35 (55)	-	-	31 (69)	-	-	0.20
Obesity, *n* (%)	25 (39)	-	-	21 (47)	-	-	0.57
Diabetes Mellitus, *n* (%)	3 (5)	-	-	4 (9)	-	-	0.37
Hypertension, *n* (%)	0	-	-	5 (11)	-	-	0.01
*Anthropometric and clinical parameters*							
Breathing frequency, ($\bar{x}$ ± SD)	15 ± 3	18 ± 10	0.03	16 ± 3	20 ± 13	0.13	0.16
Temperature, °C ($\bar{x}$ ± SD)	36	36 ± 0.6	0.20	36	36 ± 0.1	0.02	0.55
Weight, kg ($\bar{x}$ ± SD)	69 ± 12	70 ± 12	0.02	72 ± 16	73 ± 16	0.07	0.37
BMI, kg/m^2^ ($\bar{x}$ ± SD)	26 ± 4.6	27 ± 4	0.04	28 ± 5.2	27 ± 6.7	0.88	0.21
Waist, cm ($\bar{x}$ ± SD)	88 ± 10	88 ± 9	0.10	91 ± 14	91 ± 12	0.49	0.23
Hip, cm ($\bar{x}$ ± SD)	101 ± 9	102 ± 8	0.61	103 ± 11	104 ± 10	0.06	0.41
*Laboratory parameters*							
Glucose, mg/dL ($\bar{x}$ ± SD)	90 ± 11	87.7 ± 12	0.09	93 ± 16	96 ± 35	0.44	0.19
Urea, mg/dL ($\bar{x}$ ± SD)	31 ± 7	31 ± 7	0.61	33 ± 11	34 ± 9	0.62	0.18
Creatinine, mg/dL ($\bar{x}$ ± SD)	0.8 ± 0.1	0.7 ± 1	0.01	0.8 ± 0.15	0.7 ± 0.15	0.01	0.77
Uric acid, mg/dL ($\bar{x}$ ± SD)	5.1 ± 1.1	5 ± 1	0.92	5.3 ± 1.2	5.3 ± 14	0.76	0.35
Cholesterol, mg/dL ($\bar{x}$ ± SD)	181 ± 37	186 ± 36	0.21	183 ± 30	178 ± 47	0.35	0.77

At the end of the six-month follow-up, we found that the SARS-CoV-2 infection rate in our cohort of HCW was 58% (*n* = 64), and 67% (*n* = 43) of the IgG seropositive participants were asymptomatic carriers. We used paired Student’s T-test to compare numerical variables and presented data as mean ± standard deviation. We used the chi-squared test to compare categorical variables and expressed data as absolute values or percentages. We considered differences significant when *p* < 0.05. ^A^ and ^B^ indicate characteristics of IgG seropositive and seronegative participants, respectively, at the beginning of the study. ^C^ and ^D^ indicate characteristics of IgG seropositive and seronegative participants, respectively, at the end of the study. Abbreviations: IgG, immunoglobulin G; PPE, personal protection equipment; BMI, body mass index; HCW, health care workers; SARS-CoV-2, severe acute respiratory syndrome coronavirus 2; $\bar{x}$, mean; SD, standard deviation.

## Data Availability

Data are available upon request.

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
