# Peer review of "High Incidence Rate of SARS-CoV-2 Infection in Health Care Workers at a Dedicated COVID-19 Hospital: Experiences of the Pandemic from a Large Mexican Hospital"

_healthcare, 2022, doi:10.3390/healthcare10050896_

Round 1
Reviewer 1 Report
Dear Authors,
Although the work, is methodologically and scientifically acceptable, although it requires certain changes, I do not think it represents significant advances in the field.
First, it has certain limitations that should be noted. Firstly, it refers to a study of a single hospital, whose particularities, which could help to understand the results, were not described; and secondly, a low number of participants, although it is not described what percentage of the total number of hospital workers enrolled in the study, 109 seems to be a low number, with low adherence to the study. This should be clarified.
The authors established a 6-month follow-up for the subjects included in the study. Why this period of time? Are there any reference guidelines to support this choice?
Regarding table 1, the descriptive title of the table should be indicated above the table, not below it (the title of the figures is indicated below). In addition, the description between 124 and 136 is much more than a mere description, it also presents results (lines 125-126) that are not mentioned in the body of the article, and should be rewritten.
Lines 154-163. The authors point out that the incidence reported by other investigators is significantly different from that described in this work. They indicate that this may be due to the fact that those studies "use the qPCR test in either cross-sectional or prospective studies, which may lead to underestimating actual incidence rates”. However, they did not take into account that they previously state that the "HGM was "designated for the treatment of COVID-19 patients in Mexico City". Did the hospitals of the studies indicated in line 155 have the same consideration? (references, 4, 5,8 and 9)
Lines 173-176. In order to explain why the prevalence of COVID infection among workers in the different workplaces is similar, it would be interesting to know the individual protection measures, whether they were the same for all workers at the GHM.
Author Response
REVIEW REPORT REVIEWER #1
Reviewer #1
Dear Authors,
Although the work, is methodologically and scientifically acceptable, although it requires certain changes, I do not think it represents significant advances in the field.
Reply (R)
We thank you for your feedback and comments on our work. As mentioned in the manuscript, the main contribution of this work is to show that the incidence rate of SARS-CoV-2 infection is substantially high among health care professionals, who need better safety procedures in the future to face this kind of massive healthy emergency. The 2009 H1N1 pandemic and the current COVID-19 pandemic demonstrate that novel viral infections emerge and spread more rapidly than we expect, and health care workers should learn from these catastrophic events to prevent virus transmissibility and mortality rates impacting this sector. Furthermore, our findings also highlight the importance of using anti-SARS-CoV-2 IgG antibody serum titers and not only qPCR tests as a reliable methodology to monitor infection rates in health care personnel.
Query (Q) 1. First, it has certain limitations that should be noted. Firstly, it refers to a study of a single hospital, whose particularities, which could help to understand the results, were not described.
R1. The General Hospital of Mexico is one of the largest tertiary care centers in Mexico, attending nearly 200,000 low and middle-income patients from the south-central region of Mexico per year. The number of workers employed at the General Hospital of Mexico is approximately seven thousand. Health care professionals working as senior doctors, nurses, and residents are 950, 1380, and 710, respectively, divided into 48 medical specialties. In March 2020, the Ministry of Health switched the General Hospital of Mexico from a tertiary care center to a dedicated COVID-19 hospital, receiving around 60 suspected or confirmed COVID-19 cases per day, on average, during the most pronounced peaks of contagion. The number of COVID-19 patients attended during the four waves of contagion placed the General Hospital of Mexico among the five care centers with the most COVID-19 cases in the whole country. Our study was indeed conducted in one hospital; however, the hospital’s specific characteristics mentioned above made it a representative COVID-19 care center to examine and monitor the SARS-CoV-2 infection incidence in health care providers. We added the information described above in the discussion section following the Reviewer's observation. Please find this change marked yellow on pages 6 and 7.
Q2. And secondly, a low number of participants, although it is not described what percentage of the total number of hospital workers enrolled in the study, 109 seems to be a low number, with low adherence to the study. This should be clarified.
R2. The sample size (n = 109) could appear low considering more than seven thousand employees work in the General Hospital of Mexico. However, besides sample size estimation suggesting that 105 participants were enough for statistical analysis, the number of HCW enrolled allowed us to monitor any suggestive COVID-19 symptoms individually for six months, including weekends, strengthening our ability to estimate infection rates and identify asymptomatic carriers. We followed up with all participants enrolled in the study, who agreed to receive weekly phone calls for six months and attend appointments for medical consultation, clinical evaluation, and blood sampling. All participants met the criteria described above, and we can consider the adherence to the study was not low but more than acceptable by standing at 100%. Please find this information marked light green on page 3. Still, we believe your observation is of great relevance and included the sample size issue as a limitation of the study that you will be able to find marked yellow on pages 6 and 7 for clarification.
Q3. The authors established a 6-month follow-up for the subjects included in the study. Why this period of time? Are there any reference guidelines to support this choice?
R3. Yes, we chose a six-month follow-up based on previous studies assessing the COVID-19 incidence in HCW (Damluji AA et al., Seropositivity of COVID-19 among asymptomatic healthcare workers: A multi-site prospective cohort study from Northern Virginia, United States. Lancet Reg Health Am. 2021;2:100030, and Barrett ES et al., Prevalence of SARS-CoV-2 infection in previously undiagnosed health care workers in New Jersey, at the onset of the U.S. COVID-19 pandemic. BMC Infect Dis. 2020;20(1):853). We added a few lines following your observation clarifying the rationale behind choosing a six-month follow-up for this study. Please find this information marked turquoise on page 3.
Q4. Regarding table 1, the descriptive title of the table should be indicated above the table, not below it (the title of the figures is indicated below).
R4. We relocated the descriptive title of Table 1 and Figure 1 following your criticism. Please find these changes marked pink on pages 3 and 5.
Q5. In addition, the description between 124 and 136 is much more than a mere description, it also presents results (lines 125-126) that are not mentioned in the body of the article, and should be rewritten.
R5. According to your suggestion, we corrected the mistake and rewrote the description of Table 1, deleting redundant information and relocating it to the result section. Please find this change marked blue on pages 4 and 5.
Q6. Lines 154-163. The authors point out that the incidence reported by other investigators is significantly different from that described in this work. They indicate that this may be due to the fact that those studies "use the qPCR test in either cross-sectional or prospective studies, which may lead to underestimating actual incidence rates”. However, they did not take into account that they previously state that the "HGM was "designated for the treatment of COVID-19 patients in Mexico City". Did the hospitals of the studies indicated in line 155 have the same consideration? (references, 4, 5,8 and 9)
R6. As far as we know, neither of the hospitals mentioned above were switched into dedicated COVID-19 care centers, as occurred with the General Hospital of Mexico. We agree with you that admitting only COVID-19 patients for an extended period may increase the infection rates in our population of health care professionals compared to hospitals that admitted patients with other diagnoses different from COVID-19. For this reason, we added a paragraph pointing out the feasibility that converting the hospital into an entirely dedicated COVID-19 care center may impact the infection rate we found. Please find this change marked red on page 6.
Q7. Lines 173-176. In order to explain why the prevalence of COVID infection among workers in the different workplaces is similar, it would be interesting to know the individual protection measures, whether they were the same for all workers at the GHM.
R7. We provided all HCW enrolled in the protocol with the same N95 respirators, disposable surgical gowns, disposable surgical caps, and face shield masks as personal protection equipment (PPE) all along with the study. We also regularly confirmed that all HCWs correctly wore PPE independently of working in low, moderate, or high SARS-CoV-2 exposure areas within the hospital. We believe your observation is of great relevance and added information to clarify that all health care professionals enrolled in the study correctly wore the same PPE during the study, irrespectively of working in low, moderate, or high SARS-CoV-2 exposure areas. Please find this information marked teal on pages 3 and 6.
We thank you for all your comments, suggestions, and observations that have indubitably improved the last version of the manuscript.

Reviewer 2 Report
Bueno-Hernández et al. submitted article titled: „High Incidence Rate of SARS-CoV-2 Infection in Health Care Workers at a Dedicated COVID-19 Hospital: Experiences of the Pandemic from a Large Mexican Hospital”. The study is interesting and well-written but there are a few ambiguities that need to be resolved.
- Your Introduction section needs extending and further explaining what is the idea and aims behind this study. Moreover, a bit more background regarding the previous similar studies is needed.
- You should refrain from making hard statements such as „establishment of the incidence of SARS-CoV-2 infection among health care workers“. First of all, this is a single center cohort study. Secondly, you conducted this study on a very small sample hence add the term „study sample“ not just „health care workers“. Even though this study has its own value, you can’t make any strong conclusions.
- Since it is possible to find out the exact number of people which work at your hospital, it would be great if you add sample size analyses (proportions).
- You are missing the Limitations section/subsection.
Author Response
REVIEW REPORT REVIEWER #2
Reviewer #2
Bueno-Hernández et al. submitted article titled: „High Incidence Rate of SARS-CoV-2 Infection in Health Care Workers at a Dedicated COVID-19 Hospital: Experiences of the Pandemic from a Large Mexican Hospital”. The study is interesting and well-written but there are a few ambiguities that need to be resolved.
Reply (R)
We thank you for your kind comments on our work. We tried to solve all ambiguities raised on the previous version of the manuscript.
Query (Q) 1. Your Introduction section needs extending and further explaining what is the idea and aims behind this study. Moreover, a bit more background regarding the previous similar studies is needed.
R1. According to your criticism, we expanded the introduction section, incorporating more similar reports and clarifying the advantages of conducting a prospective, longitudinal protocol with measurement of specific IgG serum antibodies against the SARS-CoV-2 to estimate the infection rate in our study sample. We think the idea and aims behind the study are more precise now. Please find these changes marked dark green on page 2.
Q2. You should refrain from making hard statements such as „establishment of the incidence of SARS-CoV-2 infection among health care workers“. First of all, this is a single center cohort study. Secondly, you conducted this study on a very small sample hence add the term „study sample“ not just „health care workers“. Even though this study has its own value, you can’t make any strong conclusions.
R2. We agree with your criticism. Following your observation, we deleted any hard statement that could potentially extrapolate our findings to all health care settings. Instead, we included the sentences “in our cohort of HCW”, “in our group of HCW”, or “in our study sample of HCW” when referring to the infection rates we estimated in our study sample. Please find these changes marked purple all along with the text manuscript, lines 33, 39, 75, 77, 166, 167, 192, 198, 211, and 248.
Q3. Since it is possible to find out the exact number of people which work at your hospital, it would be great if you add sample size analyses (proportions).
R3. The number of workers employed at the General Hospital of Mexico is approximately seven thousand. At the pandemic’s beginning, two thousand aged workers were instructed to lock down at home. Then, we initially estimated a sample size considering that the number of active personnel was five thousand. Based on previous studies (all cited in the text manuscript), we expected that 13% of workers were at higher risk of SARS-CoV-2 infection (n = 650). Using the equation for population proportion n = z2(p*q)/e2 + (z2(p*q))/N, where n = sample size, z = confidence level, p = population proportion with the specific characteristic, q = population proportion without the specific characteristic, e = margin of error, and N = population size, with margin of error of 10% and confidence level of 99% we obtained n = 84. We considered 25% losses and finally found a sample size of n = 105 health care workers. We have added a sample size estimation in the manuscript following your suggestion. Please find these changes marked dark red on page 3.
Q4. You are missing the Limitations section/subsection.
R4. We have added limitations to the study following your observation. Please find these changes marked yellow on pages 6 and 7.
We thank you for all your very constructive comments on our work. Your suggestions and observations have indubitably improved the last version of the manuscript.

Reviewer 3 Report
It is an interesting and well written article, direct and easy to understand
Do participates have received any vaccination against Covid 19, is that an exclusion criteria.
How many health and staff personals has your hospital, to have an idea of the representation of 109 participants?
Participants have a PCR and a IgM/IgG test at the beginning of and at the end of the six months, do you labelled as positive if any test is positive?
How many administrative staff or with low risk are involved, perhaps this could explain the no statistical difference among health workers?
Do any participant have severe symptoms of Covid?
The prevalence of COVID-19 infection is high? Was bio protections measures assumed by all participants?
In conclusion health “There were no risk factors contributing to the increase in the infection rate, not even the workplace within the hospital” but 58% of infections is far above the average in general population or in other non-medical work places
How was R0 in general population, at that time?
Perhaps meanwhile the entire population was on lock down health, professionals are exposed to Covid ? that could explain higher rates in health personal ?
Author Response
REVIEW REPORT REVIEWER #3
Reviewer #3
It is an interesting and well written article, direct and easy to understand.
Reply (R)
We thank you for your kind comments on our work.
Query (Q) 1. Do participates have received any vaccination against Covid 19, is that an exclusion criteria.
R1. Participants enrolled in the study did not receive any vaccine against COVID-19 all along with the investigation. We added this information in the Material and Method section for clarification. Please find this statement marked grey on page 2.
Q2. How many health and staff personals has your hospital, to have an idea of the representation of 109 participants?
R2. The number of workers employed at the General Hospital of Mexico is approximately seven thousand. At the pandemic’s beginning, two thousand aged workers were instructed to lock down at home. We then considered that the number of active personnel was five thousand. Health care professionals working as senior doctors, nurses, and residents are 950, 1380, and 710, respectively, divided into 48 medical specialties. Personnel working as researchers, clinical lab technicians, psychologists, or rehabilitators are 120, 640, 85, and 310. Administrative personnel ascends to 800, among which we can find receptionists, secretaries, social workers, medical billing specialists, and schedulers, among others. For sample size estimation, we used the equation for population proportion n = z2(p*q)/e2 + (z2(p*q))/N, considering that 13% of health care personnel were at higher risk of SARS-CoV-2 infection (n = 650). With a margin of error of 10% and a confidence level of 99%, we obtained n = 84 plus 25% losses to finally found a sample size of n = 105 health care workers as the minimal for statistical analyses. So, n = 109 participants were thought of as representing the personnel at higher SARS-CoV-2 infection risk. Please find the sample size estimation and the rationale behind using n = 109 participants marked dark red on page 3.
Q3. Participants have a PCR and a IgM/IgG test at the beginning of and at the end of the six months, do you labelled as positive if any test is positive?
R3. We labeled a participant as positive for SARS-CoV-2 exposure if the IgG test was positive at the end of the study. Having either qPCR or IgG test positive at the beginning of the study was considered an exclusion criterion. Of note, all participants with positive qPCR test also showed positive IgG test at the end of the study. By labeling a participant as positive using the IgG test, we considered symptomatic subjects and asymptomatic carriers who did not request qPCR because of the absence of any symptoms concurring with COVID-19. We added this information to the manuscript text for clarification. Please find this change marked yellow on page 3.
Q4. How many administrative staff or with low risk are involved, perhaps this could explain the no statistical difference among health workers?
R4. Ten administrative staff participated in the study. Only five workers in this group were exposed to the virus by showing a positive IgG test at the end of the study. The remaining 5 participants with low exposure risk showed negative IgG test at the end of the six-month follow-up. You can find this information in Table 1 and the result section.
Q5. Do any participant have severe symptoms of Covid?
R5. Fortunately, most COVID-19 positive participants developed mild-to-moderate self-limiting symptoms, such as fever, headache, fatigue, body pain, anosmia, and dysgeusia. However, three COVID-19 positive participants developed severe symptoms, including respiratory distress (37 breaths per minute on average) and hypoxia (peripheral oxygen saturation = 78% on average), thus needing high-flow nasal oxygen as noninvasive respiratory support. You can consult this information in lines 155-159 on page 4.
Q6. The prevalence of COVID-19 infection is high? Was bio protections measures assumed by all participants?
R6. Yes. We provided all participants enrolled in the protocol with the same N95 respirators, disposable surgical gowns, disposable surgical caps, and face shield masks throughout the study. We also confirmed that all participants correctly wore the personal protection equipment (PPE) despite working in low, moderate, or high viral exposure areas within the hospital. You can see this information marked dark green on page 3.
Q7. In conclusion health “There were no risk factors contributing to the increase in the infection rate, not even the workplace within the hospital” but 58% of infections is far above the average in general population or in other non-medical work places
R7. We agree that the SARS-CoV-2 infection rate we found in our cohort of HCW was considerably higher than that observed in the community or non-medical settings. As mentioned in the manuscript, our findings align with other reports indicating that health care professionals are at much higher risk of SARS-CoV-2 infection than the general population. We found that neither comorbidities nor the workplace within the hospital contributed to the risk of infection when participants of the study correctly wore PPE. Hence, we attributed the high SARS-CoV-2 infection rate to feasibilities such as the hospital's conversion into an entirely dedicated COVID-19 care center, community exposure, and contact with family members who are asymptomatic carriers. In the discussion section, please consult these arguments, lines 205-208, 211-213, 217-219, and 221-224.
Q8. How was R0 in general population, at that time?
R8. In Mexico City, the R0 value was 1.14, on average, during the second semester of 2020. Having an R0 value with a clear growing tendency supports the idea that community exposure could considerably contribute to the high SARS-CoV-2 infection rate we observed in our cohort of HCW.
Q9. Perhaps meanwhile the entire population was on lock down health, professionals are exposed to Covid ? that could explain higher rates in health personal ?
R9. We agree that despite the lockdown reduced R0 in the general population, the risk of infection in HCW who attended COVID-19 patients for long periods kept considerably high. However, we still believe that the hospital's conversion into an entirely dedicated COVID-19 care center, community exposure, and contact with asymptomatic carriers are the main factors that contributed to the high infection rates in health care personnel we followed up.
We thank you for your very constructive comments on our work. Your suggestions and observations have indubitably improved the last version of the manuscript.

Round 2
Reviewer 1 Report
Thank you for considering the suggestions made in the first revision round.